# The Regulatory Hub of Siderophore Biosynthesis in the Phytopathogenic Fungus *Alternaria alternata*

**DOI:** 10.3390/jof9040427

**Published:** 2023-03-29

**Authors:** Je-Jia Wu, Pei-Ching Wu, Jonar I. Yago, Kuang-Ren Chung

**Affiliations:** 1Department of Plant Pathology, College of Agriculture and Natural Resources, National Chung Hsing University, Taichung 40227, Taiwan; jejiawu@gmail.com (J.-J.W.); jane871057@gmail.com (P.-C.W.); 2Plant Science Department, College of Agriculture, Nueva Vizcaya State University, Nueva Vizcaya 3700, Philippines; jyago2002@yahoo.com

**Keywords:** feedback inhibition, iron homeostasis, osmotic stress, siderophores, virulence

## Abstract

A GATA zinc finger-containing repressor (AaSreA) suppresses siderophore biosynthesis in the phytopathogenic fungus *Alternaria alternata* under iron-replete conditions. In this study, targeted gene deletion revealed two bZIP-containing transcription factors (AaHapX and AaAtf1) and three CCAAT-binding proteins (AaHapB, AaHapC, and AaHapE) that positively regulate gene expression in siderophore production. This is a novel phenotype regarding Atf1 and siderophore biosynthesis. Quantitative RT-PCR analyses revealed that only *AaHapX* and *AaSreA* were regulated by iron. AaSreA and AaHapX form a transcriptional feedback negative loop to regulate iron acquisition in response to the availability of environmental iron. Under iron-limited conditions, AaAtf1 enhanced the expression of *AaNps6*, thus playing a positive role in siderophore production. However, under nutrient-rich conditions, AaAtf1 plays a negative role in resistance to sugar-induced osmotic stress, and AaHapX plays a negative role in resistance to salt-induced osmotic stress. Virulence assays performed on detached citrus leaves revealed that AaHapX and AaAtf1 play no role in fungal pathogenicity. However, fungal strains carrying the *AaHapB*, *AaHapC*, or *AaHapE* deletion failed to incite necrotic lesions, likely due to severe growth deficiency. Our results revealed that siderophore biosynthesis and iron homeostasis are regulated by a well-organized network in *A. alternata*.

## 1. Introduction

Iron, often functioning as a cofactor of enzymes, is required for a variety of cellular processes, including deoxyribonucleotide synthesis, oxidative phosphorylation, activities of several enzymes, and detoxification of hydroxyl radicals generated from the Haber-Weiss/Fenton pathway [1,2]. However, excess iron in cells could cause cytotoxicity. Despite the abundance of total iron in the environment, most iron sources are insoluble ferric oxyhy-droxides, which are inaccessible to microorganisms [3]. It is estimated that less than 10−18 M iron in the environment is bioavailable to microorganisms [4]. Thus, microorganisms must have sophisticated mechanisms to acquire iron effectively from their environment.

Fungi have evolved three strategies: acidification of the environment, reduction of ferric iron (Fe^3+^) to a more soluble ferrous form (Fe^2+^) via the reductive iron assimilation (RIA) systems, and secretion of soluble iron-chelators called siderophores to acquire environmental iron [5,6]. The siderophore-Fe^3+^ complex is delivered into cells via specific transporters [7]. In fungal cells, ferric iron can be stored in the vacuoles and as the siderophore-Fe^3+^ complex in the cytoplasm [5,8]. The siderophore-mediated iron acquisition has been shown to be required for virulence in many pathogenic fungi of humans and plants [9,10,11].

Most fungi produce hydroxamate-type siderophores, classified into four groups: rhodotorulic acid, fusarinines, coprogens, and ferrichromes [12]. Fungal siderophores are synthesized from L-ornithine catalyzed in order by ornithine-*N^5^*-monooxygenase, transacylase, and nonribosomal peptide synthetase [5]. Deleting the nonribosomal peptide synthetase-coding gene (*Nps6*) severely impacts siderophore production and virulence in several plant pathogenic fungi, including *Cochliobolus carbonum*, *C. miyabeanus*, *Fusarium graminearum*, and *Alternaria brassicicola* [13], indicating the important role of siderophore-mediated iron acquisition in fungal invasion to their respective host plants.

Siderophore biosynthesis and iron acquisition must be tightly regulated to maintain iron homeostasis and avoid iron toxicity. *Saccharomyces cerevisiae* utilizes two transcription factors, Aft1 and Aft2, to regulate iron uptake, recycling, and mobilization [14,15,16]. Filamentous fungi utilize a GATA repressor (SreA) and a basic leucine zipper (bZIP)-containing regulator (HapX) to control the expression of iron-acquisition genes and to maintain siderophore production and iron homeostasis [17,18]. HapX has been shown to interact with HapB and be recruited to bind to a CCAAT-binding complex (CBC) composed of three proteins, HapB, HapC, and HapE, in *Aspergillus nidulans* [19,20]. Under iron-depleted conditions, HapX promotes siderophore production and iron acquisition by suppressing the expression of *SreA* [17]. HapX is also known to suppress the genes involved in iron consumption, including heme cytochrome C, iron-sulfur cluster-containing aconitase, and homoaconitase under iron-depleted conditions [21,22,23,24]. In contrast, under iron-replete conditions, SreA suppresses the expression of *HapX* and many genes involved in siderophore biosynthesis and iron uptake [25]. Although both SreA and HapX are required to regulate siderophore biosynthesis, their roles in fungal virulence vary in different fungal species. SreA is not required for virulence in the human pathogens *fumigatus* [26] and *Candida albicans* [27] but is required for full virulence in the human pathogen *Cryptococcus neoformans* [28] and the maize pathogen *C. heterostrophus* [29]. HapX is required for virulence in *A. fumigatus*, *C. albicans*, and *C. neoformans* [22,27,30], as well as the plant pathogenic fungi *Verticillium dahlia* [24] and *F. oxysporum* [23].

The tangerine pathotype of *A. alternata* produces a host-selective toxin called ACT to kill susceptible citrus cultivars before colonization and acquire nutrients from dead tissues [31]. ACT is the primary pathogenicity determinant as the ∆*AaACTT6* mutation strain fails to accumulate ACT and induce necrotic lesions in the susceptible citrus cultivar [32]. In addition, the detoxification of toxic reactive oxygen species (ROS) and the production of cutinases and siderophores are required for virulence [10,32,33]. During the invasion, *A. alternata* must overcome the toxicity of host-generated ROS and acquire iron from host tissues to establish successful colonization. Iron can facilitate the detoxification of hydrogen peroxide (H_2_O_2_), promote the activity of antioxidant enzymes (e.g., catalase and superoxide dismutase), and assist fungal invasion [34]. Deleting an *AaNps6* gene in the tangerine pathotype of *A. alternata* results in a mutant that fails to synthesize siderophores, increases sensitivity to H_2_O_2_, and reduces virulence in citrus [10], indicating that siderophore- mediated iron uptake plays a critical role in *A. alternata* pathogenesis.

*A. alternata* produces coprogens and hydroxycoprogens [35]. We have previously demonstrated that the biosynthesis of siderophores is negatively regulated by AaSreA under iron-replete conditions [36]. However, the AaSreA-deficient mutant displays wild-type virulence in citrus. To better understand the regulatory pathways controlling the biosynthesis of siderophores and iron uptake, we investigated the functions of AaHapX and three CCAAT-binding proteins (AaHapB, AaHapC, and AaHapE) in siderophore production, stress resistance, and virulence in *A. alternata*. Moreover, a yeast-activating transcription factor 1 (Atf1) homolog containing a bZIP domain was found to play a role in siderophore biosynthesis primarily by activating the expression of *AaNps6*. AaAtf1 was misidentified from the genome of *A. alternata* when searching for the yeast Aft1 and Aft2 homologs. Transcriptional interactions among these regulators under iron-depleted and replete conditions were established. This study demonstrated a complex regulatory network, directly and indirectly controlling siderophore biosynthesis and iron acquisition in the phytopathogenic fungus *A. alternata*.

## 2. Materials and Methods

### 2.1. Fungal Strains and Growth Conditions

The wild-type EV-MIL-31 strain of *A. alternata* was isolated from a diseased leaf of Minneola tangelo (*Citrus* x *tangelo* J.W. Ingram & H.E. Moore) and used in this study as a recipient host for transformation and the mutagenesis experiments. The *AaSreA*-deficient mutants (∆*AaSreA*_D6 and D12) carrying a deletion of the gene encoding a GATA-type transcription suppressor were obtained from a separate study [36]. ∆*AaNps6*, a defective strain for the biosynthesis of siderophores, was generated by Chen et al. (2013) [10]. ∆*AaACTT6* carrying a mutation of the ACT biosynthetic gene, *AaACTT6*, was generated by Ma et al. (2019) [32]. Unless otherwise stated, fungal strains were grown on potato dextrose agar (PDA; Difco, Sparks, MD, USA) or minimal medium (MM) [36] at 28 °C for 3 to 5 days. For the isolation of nucleic acids, fungi were grown in a complete medium (MM+ 1 g/L yeast extract and 1 g/L casein hydrolysate). For ACT toxin production, fungi were grown in a modified Richard’s medium [37] for 21 days. A regeneration medium (RMM) [38] was used to recover fungal transformants. For medium shift experiments, fungal strains were grown in PDB for 2 to 3 days. Mycelium was collected through a sterile filter paper, washed with sterile water, transferred to MM or MM amended with an appropriate compound, and incubated for an additional 24 h.

### 2.2. Sensitivity Assays

Cellular sensitivity to chemicals was carried out on PDA or MM by transferring fungal mycelia as a pipette-tip (10 µL) inoculation to a medium containing a test compound at an appropriate concentration. Chemicals used for sensitivity tests included ferric chloride (FeCl_3_, 0.2 mM), bathophenanthrolinedisulfonic acid (BPS, an iron chelator, 0.2 mM), hydrogen peroxide (H_2_O_2_, 5 to 15 mM), *tert*-butyl-hydroperoxide (tBHP, 1.5 and 3.75 mM), menadione (MD, 0.1 and 0.2 mM dissolved in dimethyl sulfoxide (DMSO)), diethyl malonate (DEM, 2.5 mM dissolved in methanol (MeOH)), glucose (1 M), sucrose (1 M), mannitol (1 M), sorbitol (1 M), sodium chloride (NaCl, 1 M), potassium chloride (KCl, 1 M), and Congo red (CR, 150 µM dissolved in ethanol (EtOH)). Unless otherwise indicated, all chemicals were dissolved in Milli-Q water. The percentage of growth inhibition was calculated by dividing the relative difference in the growth by the wild-type growth and multiplying by 100.

### 2.3. Targeted Gene Deletion and Genetic Complementation in A. alternata

A split marker-mediated transformation [38] was used to delete *AaHapX*, *AaHapB*, *AaHapC*, *AaHapE*, and *AaAtf1* in *A. alternata*. A hygromycin phosphotransferase gene cassette (*HYG*, 2.5 kb) was split into two overlapping fragments HYg and hYG, using PCR with primer pairs M13F/hyg4 and hyg3/M13R, respectively, from an *HYG*-containing plasmid. Two split marker fragments, HYg::5′ target gene DNA fragment and hYG::3′ target gene fragment, were generated by two-step fusion PCR with gene-specific primers (Appendix A Appendix A) and directly transformed into protoplasts prepared from wild type using polyethylene glycol and CaCl_2_ as previously described [39]. Fungal transformants were recovered from RMM amended with 200 µg/mL hygromycin (Roche Applied Science, Indianapolis, IN, USA), examined by PCR with a gene-specific primer pairing with a *HYG* primer (hyg3 or hyg4), and verified further by Southern blot analyses using gene-specific and *HYG* probes or PCR-restriction enzyme length polymorphism (PCR-RFLP) patterns (Appendix A Appendix A). Two independent mutants from each gene deletion were used for phenotypic analyses. Defective phenotypes were restored by transforming a functional copy of the gene into a deletion mutant. Each gene, including its endogenous promoter, was amplified by PCR with gene-specific primers and cloned into pCB1532 carrying a sulfonylurea-resistance (*Sur*) gene [40]. To facilitate cloning, recognition sequences of restriction endonucleases were incorporated into the primers. Fungal transformants were selected on RMM amended with 5 g/mL sulfonylurea (Chem Service, West Chester, PA, USA) and tested for restoration of radial growth on PDA or siderophore production. Oligonucleotide primers used for PCR and Southern blot analyses are listed in Appendix A Appendix A.

### 2.4. Miscellaneous Procedures for Manipulation of Nucleic Acids

Fungal genomic DNA was obtained using a Genomic DNA Mini Purification kit (BioKit, Taipei, Taiwan) or a phenol/chloroform DNA extraction protocol. Fungal RNA was isolated using TRI reagent (Sigma-Aldrich, St. Louis, MO, USA) and purified further using PureLink RNA Mini Kit (Invitrogen, Carlsbad, CA, USA) following the manufacturer’s instructions. DNA digestion with restriction endonucleases, electrophoresis, DNA ligation, bacterial transformation, Southern blot hybridizations, and post-hybridization washing were carried out according to standard procedures. Digoxigenin (DIG)-11-dUTP (Roche Applied Science, Indianapolis, IN, USA) was incorporated by PCR into a DNA fragment with the gene-specific primers and used for Southern blot analyses. The probe was detected by an immunological assay according to the manufacturer’s instructions (Roche Applied Science, Indianapolis, IN, USA). The protein-coding genes were predicted by GlimmerHMM [41,42]. Pairwise sequence comparison was performed using CLC Genomic Workbench 9.5.1 (CLC Bio, Qiagen, Aarhus, Denmark) to calculate genetic distance using the Jukes–Cantor model and percent identity. Conserved domains were predicted with the CD-search tool [43] available in the National Center for Biotechnology Information (NCBI) and the MEME Suite server [44]. 

### 2.5. Assays for Siderophore Production

The production of siderophores was assayed on a Chrome azurol S (CAS)-containing medium as described [45]. The formation of orange halos on the blue background around the fungal colony indicated the production of siderophores. The plates were photographed, and the radius of orange halos was determined using Image J software V. 1.54b (US National Institutes of Health, Bethesda, MD, USA). Siderophores were further isolated from culture filtrates of fungal strains cultured in MM for 5 to 7 days with Amberlite XAD-16 resin (Sigma-Aldrich, St. Louis, MO, USA) and examined by thin-layer chromatography (TLC) and high-performance liquid chromatography (HPLC) using mobile phases and conditions as described [35].

### 2.6. Quantitative RT-PCR and Gene Expression Analyses

Gene expression was evaluated by quantitative real-time PCR (qRT-PCR) using gene-specific primers (Appendix A Appendix A). All qRT-PCR reactions were set up using iQ SYBR Green Supermix (Bio-Rad, Hercules, CA, USA) and performed in a CFX Connect model of Real-Time PCR Detection System (Bio-Rad, Hercules, CA, USA). Amplification of the *β*-tubulin coding gene was included as an internal control, and the specific transcripts were assessed by the melting curve. The relative expression level was determined by a comparative Ct method (∆∆CT). All treatments were conducted with three biological replicates, and the significant difference was determined by statistical analysis. Experiments were repeated at least two times.

### 2.7. Assays for Virulence and Toxin Production

*A. alternata* virulence was assessed on detached leaves of calamondin, a cross between a mandarin orange (*Citrus reticulata* Blanco) and a kumquat (*Fortunella margarita* Swingle). Leaves (6 to 8 days after emergence and approximately 3 to 4 cm in length) were harvested, washed with water, and inoculated with agar plugs carrying fungal hyphae. Some leaf spots were wounded by a fine needle before inoculation. Alternatively, leaves were inoculated by placing 5 µL of conidia suspensions (2 × 10^5^ conidia/mL) on each spot. Leaves treated with water were used as mock controls. All leaves were kept in a plastic container for 3 to 5 days. ACT was purified with Amberlite XAD-2 resin (Sigma-Aldrich, St. Louis, MO, USA), separated in a silica gel 60 F254 plate (Merck KGaA, Darmstadt, Germany) using methanol/acetic acid/water (4:1:5, *v/v*) as mobile phase, visualized under a hand-held UV illuminator, and photographed.

### 2.8. Statistical Analysis

Unless stated otherwise, all experiments with multiple replicates were conducted at least two times. The significance of treatments was analyzed by one-way ANOVA and separated by Tukey’s HSD post hoc test (*p* < 0.05).

## 3. Results

### 3.1. Identification and Characterization of Five Transcription Regulators

Sequences of five DNA-binding proteins, AaHapX (528 a.a., OP828655), AaHapB (343 a.a., OP828656), AaHapC (188 a.a., OP828657), AaHapE (313 a.a., OP828658), and AaAtf1 (545 a.a., OP828654), were obtained from the complete genome sequence of *A. alternata*.

AaHapX is a bZIP protein, which contains a Hap4-like CCAAT-binding complex (CBC) domain, a coiled-coil DNA binding domain, and three cysteine-rich regions (CRR) (Figure 1). AaHapB is a CCAAT-binding transcription factor subunit B (CBF-B), also known as nuclear transcription factor Y alpha (NF-YA). AaHapC is a CCAAT-binding factor (CBF/NF-YB) belonging to a histone-like transcription factor. AaHapE is a CCAAT- binding factor subunit C belonging to the HAP5 superfamily. AaAtf1 contains an Aft1 osmotic stress response (OSM) domain, Aft1 homologous recombination activation (HRA; IPR021755) and repression (HRR; IPR021755) domains involved in meiotic recombination, a coiled-coil DNA-binding domain, and a bZIP_ATF2 domain. AaAtf1, AaHapX, AaHapB, AaHapC, and AaHapE of *A. alternata* shared low overall identities (less than 55%) with their corresponding orthologs of fungi (Appendix A Appendix A).

### 3.2. Transcription Regulators Are Required for Growth and Iron Homeostasis

Two split *HYG* marker gene fragments were generated from each of the five genes and used for targeted gene deletion in the wild-type strain of *A. alternata*. As a result, at least two mutants were identified from each gene after verification by Southern blot analyses or RFLP patterns. Only mutants carrying a single *HYG* insertion were selected for further analyses. The *AaHapX* deficient mutants (∆*AaHapX*_X6 and X10) produced fewer aerial hyphae than the wild type on PDA and MM (Figure 2). Adding BPS (an iron chelator) into PDA or MM reduced ∆*AaHapX* growth considerably. Adding FeCl_3_ into MM but not PDA also reduced ∆*AaHapX* growth. The growth of fungal strains carrying gene deletion in *AaHapB* (∆*AaHapB*_B7 and B8), *AaHapC* (∆*AaHapC*_C12 and C45), or *AaHapE* (∆*AaHapE*_E1 and E2) was severely impaired, especially on MM amended with BPS (Figure 2). Compared to the wild type, *AaAtf1* deficient mutants (∆*AaAtf1*_A2 and A3) reduced growth on PDA by 31%. In contrast, ∆*AaAtf1* grew slightly better than the wild type on MM. Adding BPS or FeCl_3_ into PDA or MM did not affect ∆*AaAtf1* growth. Compared to the wild type, the *AaSreA* deficient mutants reduced growth on PDA and MM by 80% and 40%, respectively. Adding BPS into PDA greatly enhanced ∆*AaSreA* growth, whereas adding FeCl_3_ into MM reduced ∆*AaSreA* growth. Defective growth in each gene mutation was restored to the wild-type level after re-expressing a functional copy of the gene in the respective mutant (Figure 2).

### 3.3. Biosynthesis of Siderophores Is Positively and Negatively Regulated by Multiple Transcription Factors

CAS and TLC assays of siderophores revealed that the production of siderophores was severely impaired in the ∆*AaHapX*, ∆*AaHapB*, ∆*AaHapC*, ∆*AaHapE*, and ∆*AaAtf1* strains (Figure 3). In contrast, the production of siderophores was greatly increased in ∆*AaSreA*. The complementation strains in each of the gene mutations had the wild-type level of siderophore production. Siderophores produced by the wild-type, the mutants (∆*AaHapX* and ∆*AaAtf1*), and their complementation strains were also confirmed by HPLC analyses (Appendix A Appendix A).

### 3.4. Cross-Interactions between Transcriptional Regulators under Different Iron Conditions

Quantitative RT-PCR analyses revealed that *AaHapX* was preferably expressed under iron-depleted conditions, whereas *AaSreA* was highly expressed under iron-replete conditions (Figure 4). Iron had little or no effect on the expression of *AaAtf1*, *AaHapB*, *AaHapC*, and *AaHapE* (Figure 4 and Appendix A). Deleting *AaHapX* increased the expression of *AaSreA* under iron-depleted conditions. Under iron-rich conditions, deleting *AaHapX* had little or no effect on the expression of *AaSreA*. Deleting *AaHapX* slightly increased the expression of *AaAtf1*, particularly under iron-depleted conditions. Deleting *AaSreA* significantly 245 increased the expression of *AaHapX* and *AaAtf1*, primarily under iron-replete conditions. Deleting *AaAtf1* had little or no effect on the expression of *AaHapX* and *AaSreA* under either iron-depleted or replete conditions.

### 3.5. The Genes Involved in Siderophore Biosynthesis and Iron Acquisition Are Differentially Regulated by AaHapX and AaAtf1

Under iron-depleted conditions, deleting *AaHapX* decreased the expression of *AaNps6* (encoding a nonribosomal peptide synthetase) (Figure 5). In contrast, under iron-replete conditions, deleting *AaHapX* increased the expression of *AaNps6*, *AaSit1* (encoding a siderophore iron transporter), and *AaMirB* (encoding a siderophore iron transporter). Deleting *AaAtf1* decreased the expression of *AaNps6* under iron-depleted conditions; however, this had no effect on the expression of *AaNps6*, *AaSit1*, and *AaMirB* under iron-replete conditions.

### 3.6. AaAtf1 and AaHapX Play a Negative Role in Osmotic Stress under Nutrient-Rich Conditions

Sensitivity tests assayed on PDA revealed that deleting *AaAtf1* increased resistance to high glucose, sucrose, mannitol, and sorbitol but not NaCl and KCl (Figure 6A). Compared to the wild type, ∆*AaAtf1* reduced radial growth by 31% on PDA, whereas ∆*AaAtf1* exhibited growth similar to wild type on PDA amended with a sugar osmoticant. In contrast, deleting *AaHapX* increased resistance to NaCl and KCl but not sugar osmoticants (Figure 6B). ∆*AaAtf1* showed no significant difference in growth inhibition on MM amended with or without osmoticants (Appendix A Appendix A). When assayed on PDA and MM, deleting *AaAtf1* did not affect resistance or sensitivity to H_2_O_2_, tBHP, menadione, diethyl maleate, and Congo red. Deleting *AaHapX* also unchanged sensitivity to H_2_O_2_ (Appendix A Appendix A).

### 3.7. Siderophore-Related Regulators Play No Role in Toxin Production and A. alternata Virulence

Pathogenicity assays were conducted on detached calamondin leaves, revealing that ∆*AaAtf1* and ∆*AaHapX* induced necrotic lesions at rates and magnitudes similar to the wild type 3 days post inoculation (dpi) (Figure 7A). Because ∆*AaHapB*, ∆*AaHapC*, and ∆*AaHapE* grew slowly and produced few conidia, their pathogenicity was assessed on detached leaves using mycelial mass. The results revealed that ∆*AaHapB*, ∆*AaHapC*, and ∆*AaHapE* were unable to induce necrotic lesions 3 dpi, even when the leaves were wounded before inoculation. ACT was isolated from culture filtrates of the wild-type, the ∆*AaAtf1*, and the CPA1 complementation strains and analyzed by TLC, revealing no significant difference in the level of ACT (Figure 7B). As mock controls, no ACT was detected from medium-only or culture filtrates of an ACT deficient strain (∆*AaACTT6*).

## 4. Discussion

Iron is an essential trace element required for metabolic, enzymatic, and regulatory functions in all living cells [46]. In contrast, excess iron could be toxic. Thus, all cells must have elaborate systems to maintain iron homeostasis [47,48]. Previously, we have demonstrated that, under iron-replete conditions, *A. alternata* utilizes AaSreA to suppress the production and transport of siderophores to avoid iron toxicity [36]. In the present study, we characterized five transcription regulators that positively controlled the biosynthesis of siderophores in the phytopathogenic fungus *A. alternata*. Our results indicated that *A. alternata* mainly relies on a transcriptional feedback inhibition between the AaSreA iron repressor and the AaHapX transcription factor to regulate iron acquisition (Figure 8). AaHapX mainly functions under iron-depleted conditions, whereas AaSreA functions under iron-replete conditions.

The *A. alternata SreA* encodes a polypeptide containing two conserved GATA-type zinc finger domains and plays a determinant role in the biosynthesis of siderophores [36]. The expression of *AaSreA* was upregulated under iron-rich conditions and downregulated under iron-depleted conditions. In contrast, the expression of *AaHapX* was downregulated under iron-rich conditions and upregulated under iron-depleted conditions. Thus, AaSreA serves as a repressor under iron-replete conditions, and the bZIP transcription regulator AaHapX acts as an activator under iron-depleted conditions. When environmental iron is scarce, the expression of *AaSreA* is inhibited by AaHapX (Figure 8). AaHapX may also directly upregulates the gene (*AaNps6*) involved in the biosynthesis of siderophores under iron-depleted conditions. When environmental iron is abundant, *AaSreA* is highly expressed, and the AaSreA protein suppresses the expression of the genes involved in the biosynthesis and transportation of siderophores. AaSreA also suppressed the expression of the *AaHapX* gene under iron-rich conditions. The results indicated that AaSreA and AaHapX are the major siderophore regulators by forming a negative feedback loop to 305 regulate iron acquisition in response to the availability of environmental iron in *A. alternata*. A similar transcriptional feedback inhibition between SreA and HapX has been reported in *Aspergillus nidulans* [17], confirming the important role of SreA and HapX in siderophore production in filamentous fungi.

HapX interacts with the CCAAT binding proteins (HapB, HapC, and HapE) and forms a CBC complex involved in suppressing the expression of *SreA* and activating the biosynthesis of siderophores under iron-depleted conditions in *Aspergillus* spp. [17,47,49]. The function of HapX is highly dependent on the CBC/HapX interaction [50]; however, deleting the HapX-coding gene homolog could lead to different deficiencies in different fungal species. In *A. alternata*, deleting *AaHapX* exhibited reduced siderophore production; however, this had a moderate impact on growth, whereas deleting *AaHapB*, *AaHapC*, or *AaHapE* resulted in severe growth retardation, indicating that AaHapB/C/E might also be independent of AaHapX to regulate the expression of genes required for growth in *A. alternata*. The results also suggest that AaHapX primarily responds to iron availability. In *F. oxysporum*, deleting the *HapX* homolog upregulates the genes involved in the iron-consuming pathways but has no effect on the iron acquisition, resulting in growth reduction under iron-depleted conditions and lower virulence in hosts [23]. In *V. dahlia*, HapX is required for iron homeostasis, growth, the formation of conidia and microsclerotia, resistance to H_2_O_2_, and virulence [24]. Sensitivity assays revealed that AaHapX plays no role in resistance to oxidative stress and iron resistance in *A. alternata*. The function of HapX in regulating siderophore biosynthesis and virulence has also been reported in the opportunistic pathogens *A. fumigatus*, *C. albicans*, and *C. neoformans* of humans [22,27,30] and in the insect mycopathogen *Beauveria bassiana* [51]. In contrast, deleting *AaHapX* in *A. alternata* had no impact on virulence, even though the mutant is defective in siderophore production. Deleting the AaSreA repressor-coding gene resulting in high siderophore production also has no impact on *A. alternata* virulence [36]. However, deleting the *AaNps6* gene completely blocks siderophore biosynthesis and results in lower virulence in citrus [10]. Thus, a low-level production of siderophores might be sufficient to enable *A. alternata* to colonize citrus hosts as long as toxin production is unaffected, as was revealed in the ∆*AaAtf1* mutant. ∆*AaHapB*, ∆*AaHapC*, and ∆*AaHapE* failed to produce necrotic lesions mainly due to severe growth defects. In addition to siderophore production, AaHapX confers sensitivity to salts in *A. alternata* as the ∆*AaHapX* mutant grew much better than the wild type on PDA amended with NaCl or KCl. This is a novel phenotype associated with AaHapX. Interconnection between iron homeostasis and osmotic stress response has also been reported in plants [52] and the halophilic bacterium, *Chromohalobacter salexigens* [53].

In addition to AaSreA and AaHapX/B/C/E, we found that AaAtf1 containing a bZIP_ATF2 domain at its C terminus is involved in siderophore biosynthesis. This novel phenotype regarding Atf1 and siderophore production has never been reported in any fungi. Atf1 has been shown to be involved in stress response, sexual development, and meiotic hot spot activation in the fission yeast *Schizosaccharomyces pombe* [54,55], *Cryptococcus neoformans* [56] and many filamentous fungi, including *A. nidulans* [57], *A. oryzae* [58], *N. crassa* [59], *A. fumigatus* [57], *Mucor circinelloides* [60], *Penicillium marneffei* [61], *F. graminearum* [62], *F. verticillioides* [63], *F. oxysporum* f. sp. *cubense* [64], *V. dahlia* [65], *Claviceps purpurea* [66], *Botrytis cinerea* [67], and *M. oryzae* [68]. Depending on fungal species, Atf1 plays a role in resistance to heat, cold, desiccation, cell wall-disrupting agents (Congo red and calcofluor white), fungicides (fludioxonil and caspofungin), osmotic or oxidative stress, and the maintenance of iron homeostasis [69]. AaAtf1 plays a negative role in osmotic resistance under nutrient-rich conditions, consistent with the finding that AaAtf1 contains an osmotic stress response (OSM) domain. However, AaAtf1 is not required for fungal virulence as assayed on detached citrus leaves. Sensitivity tests revealed that ∆*AaAtf1* grew much slower than wild type on PDA and displayed wild-type growth on PDA but not MM amended with glucose, sucrose, mannitol, or sorbitol. Unlike other fungal species, AaAtf1 plays no role in resistance to cell-wall-disrupting agents, heat/cold, or oxidative stress. Surprisingly, AaAtf1 is involved in siderophore biosynthesis by regulating the expression of *AaNps6*. The results indicate that Atf1 homologs evolve in different fungi, providing a better adaptation to their biological environments.

In conclusion, siderophore-mediated iron uptake is required for growth, conidiation, ROS resistance, and virulence in *A. alternata*. This fungus depends on the AaSreA repressor and the AaHapX/CBC complex activator in response to the iron availability in the environment. Siderophore biosynthesis is also regulated by AaAtf1. AaAtf1 activates the expression of *AaNps6* under iron-depleted conditions and facilitates siderophore biosynthesis. Although both AaAtf1 and AaHapX are involved in siderophore production, AaAtf1 plays a negative role in resistance to sugar-induced osmotic stress, and AaHapX plays a negative role in resistance to salt-induced osmotic stress under nutrient-rich conditions. This study further highlights that a well-regulated network is orchestrated to control the biosynthesis of siderophore, iron uptake, and stress response in *A. alternata*. Further studies using RNA sequencing analyses will allow us to identify key or novel genes that are regulated by AaHapX, AaAtf1, and AaSreA.

## Figures and Tables

**Figure 1 jof-09-00427-f001:**
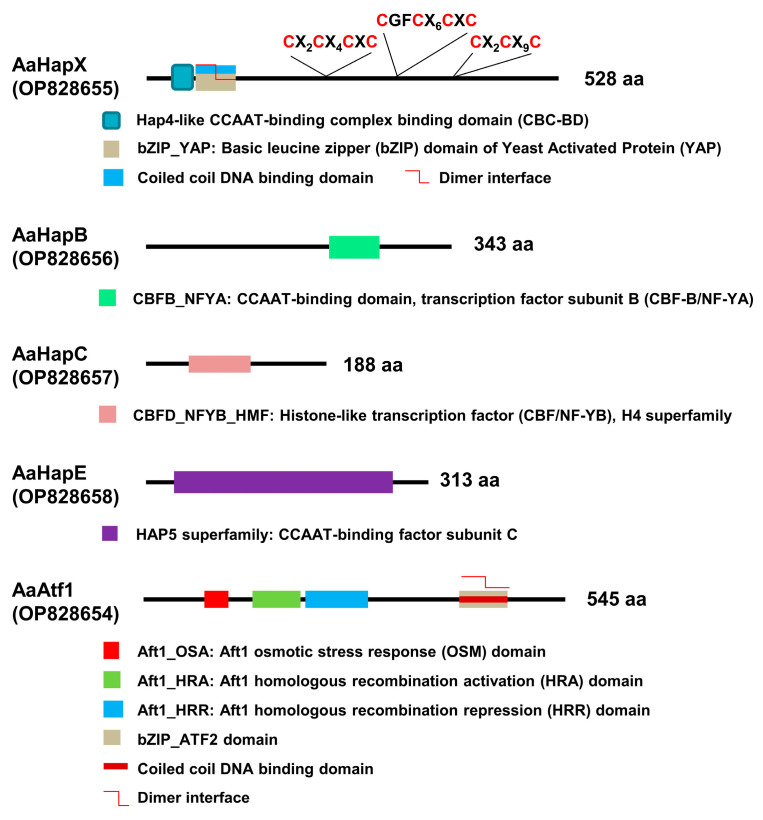
Schematics showing the functional domains of five transcription regulators associated with siderophore production in *A. alternata*. Both AaHapX and AaAtf1 are bZIP-containing transcription factors. Three cysteine-rich regions conserved in the AaHapX are also indicated. AaHapB, AaHapC, and AaHapE are CCAAT-binding proteins.

**Figure 2 jof-09-00427-f002:**
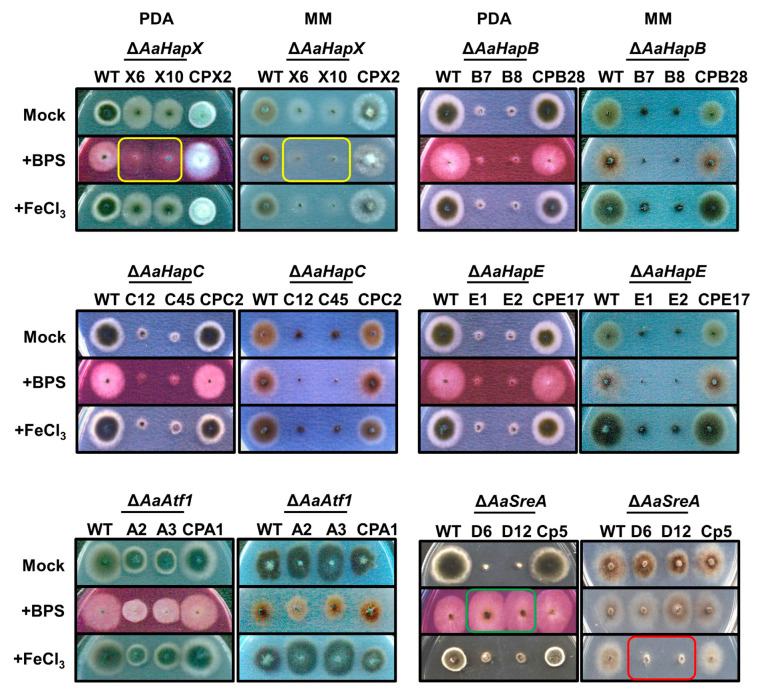
Transcription regulators involved in iron homeostasis and growth in *A. alternata* under iron-replete (PDA) and iron-depleted (MM) conditions. Fungal strains, including wild type (WT), deletion mutants (∆), and complementation strains (CP) were transferred by pipette tips to PDA and MM amended with or without 0.2 mM FeCl_3_ or 0.2 mM bathophenanthrolinedisulfonic acid (BPS) and incubated at 28 °C for 3 to 5 days. ∆*AaHapX* grew poorly on a medium amended with BPS (indicated by yellow rectangles). However, ∆*AaSreA* grew poorly under iron-rich conditions (indicated by a red rectangle), and exhibited wild-type growth in the presence of BPS (indicated by a green rectangle). ∆*AaHapB*, ∆*AaHapC*, and ∆*AaHapE* grew poorly on either medium. Experiments were tested at least two times with two replicates for each treatment.

**Figure 3 jof-09-00427-f003:**
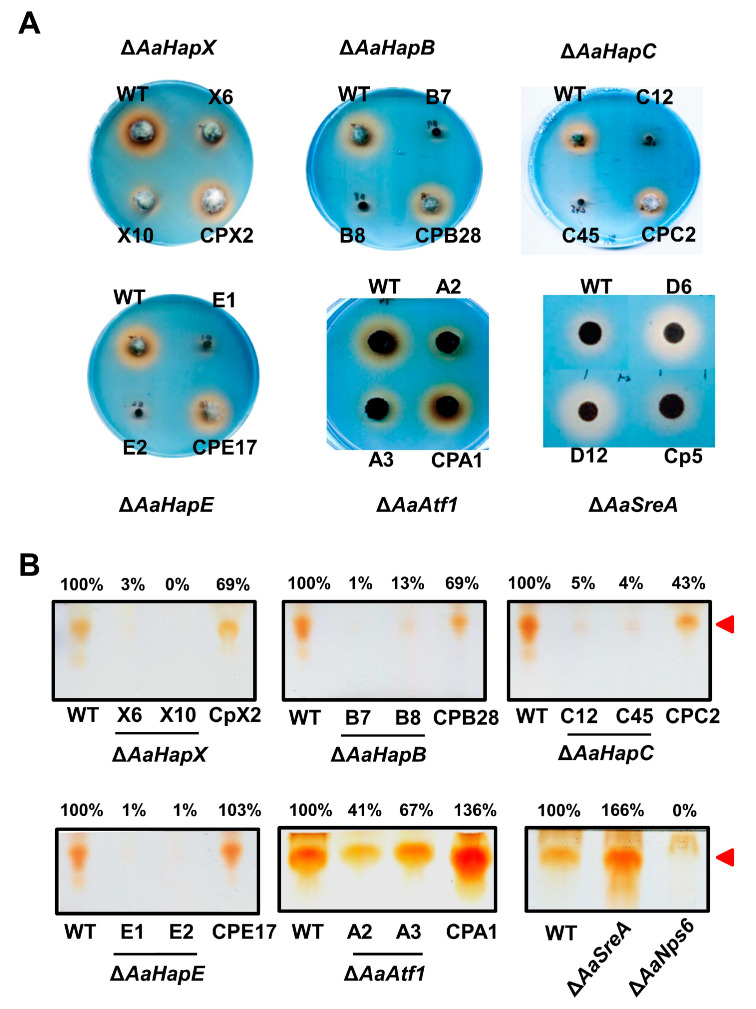
Siderophore production by *A. alternata* strains. (**A**) Fungal strains, including wild type (WT), deletion mutants (∆), and complementation strains (CP), were transferred by a pipette tip (10 µL) to chromeazurol S-containing agar medium and incubated at 28 °C for 3 to 5 days. The formation of an orange halo around the fungal colony indicates the production of siderophores. (**B**) TLC analyses of siderophores. Fungal strains were cultured in liquid MM for 5 to 7 days, and culture filtrates were mixed with FeCl_3_ and Amberlite XAD-16 resin. Siderophores were eluted from the resin with methanol and analyzed by TLC. Relative abundance of the brown spots (*Rf*∼0.75, indicated by red arrowhead) were analyzed by Image J software V 1.54b by referring to the wild type.

**Figure 4 jof-09-00427-f004:**
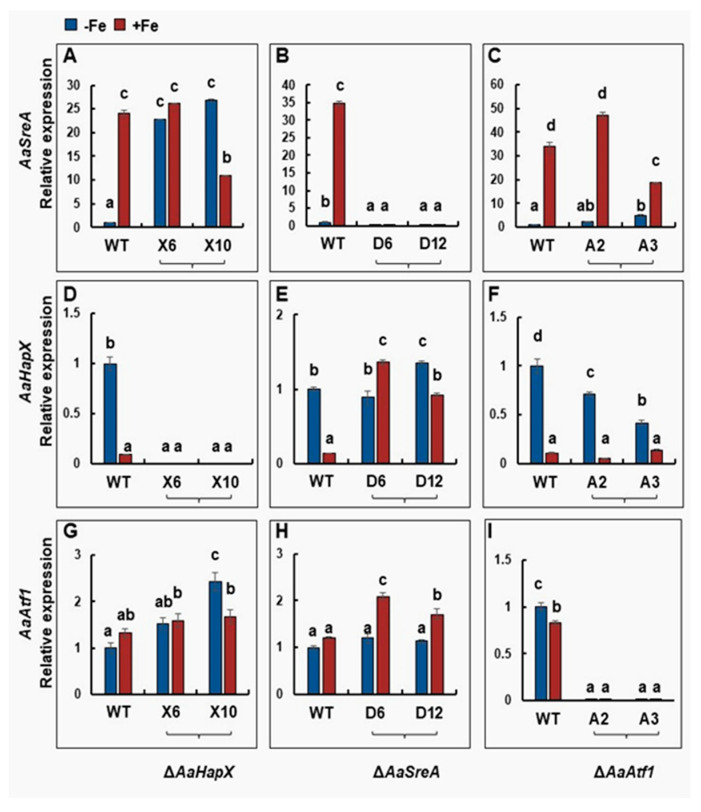
Quantitative real-time PCR analyses of the expression of the *AaSreA* (**A**–**C**), *AaHapX* (**D**–**F**), and *AaAtf1* (**G**–**I**) genes in *A. alternata* strains grown in the presence or absence of iron. Wild type, ∆*AaHapX* (X6 and X10), ∆*AaSreA* (D6 and D12), and ∆*AaAtf1* (A2 and A3) were grown in PDB at 28 °C for 2 to 3 days. Mycelium was harvested, washed with water, transferred into liquid MM or MM amended with 0.2 mM FeCl_3_, and incubated for an additional 24 h. RNA was purified from mycelium and used for first-strand cDNA synthesis. Quantitative RT-PCR was performed using gene-specific primers. The relative expression level from three independent reactions was calculated by a comparative Ct method (∆∆CT) in relation to the expression of the fungal *β*-tubulin-coding gene. In each assay, the expression level in the wild type grown in MM was set to 1. Means indicated by the same letter are not significantly different, *p* < 0.05. All experiments were repeated at least two times, showing similar trends.

**Figure 5 jof-09-00427-f005:**
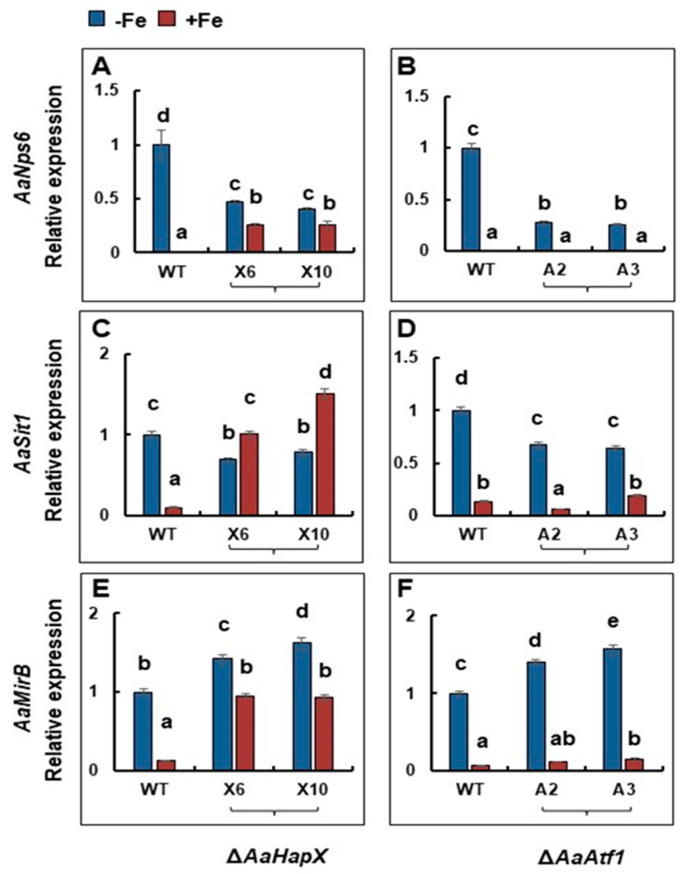
Quantitative real-time PCR analyses of the expression of siderophore biosynthesis- and transport-related genes in *A. alternata* strains grown in the presence or absence of iron. Wild type, ∆*AaHapX* (X6 and X10), and ∆*AaAtf1* (A2 and A3) were grown in PDB at 28 °C for 2 to 3 days. Mycelium was harvested, washed with water, transferred into liquid MM or MM amended with 0.2 mM FeCl_3_, and incubated for an additional 24 h. RNA was purified from mycelium and used for first-strand cDNA synthesis. Quantitative RT-PCR was performed using primers targeting *AaNps6* (**A**,**B**), *AaSit1* (**C**,**D**), or *AaMirB* (**E**,**F**). The relative expression level from three independent reactions was calculated by a comparative Ct method (∆∆CT) in relation to the expression of the fungal *β*-tubulin-coding gene. In each assay, the expression level in the wild type grown in MM was set to 1. Means indicated by the same letter are not significantly different, *p* < 0.05. All experiments were repeated at least two times, showing similar trends.

**Figure 6 jof-09-00427-f006:**
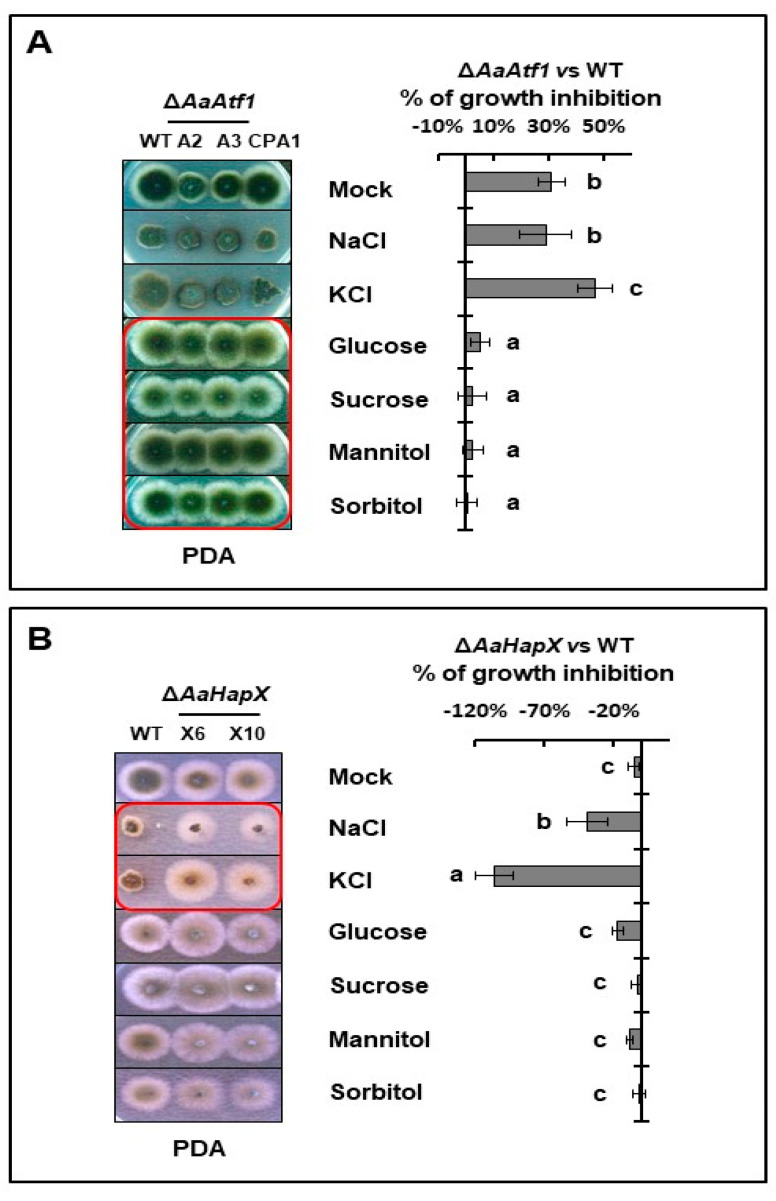
AaAtf1 and AaHapX play different roles in stress resistance. (**A**) Sensitivity tests assayed on PDA revealed that, compared to wild type, ∆*AaAtf1* increased resistance to glucose, sucrose, mannitol, and sorbitol. (**B**) ∆*AaHapX* increased resistance to KCl and NaCl. The percent of growth inhibition (%) is shown on the right side. Means indicated by the same letter are not significantly different, *p* < 0.05. All experiments were repeated at least two times, showing similar trends.

**Figure 7 jof-09-00427-f007:**
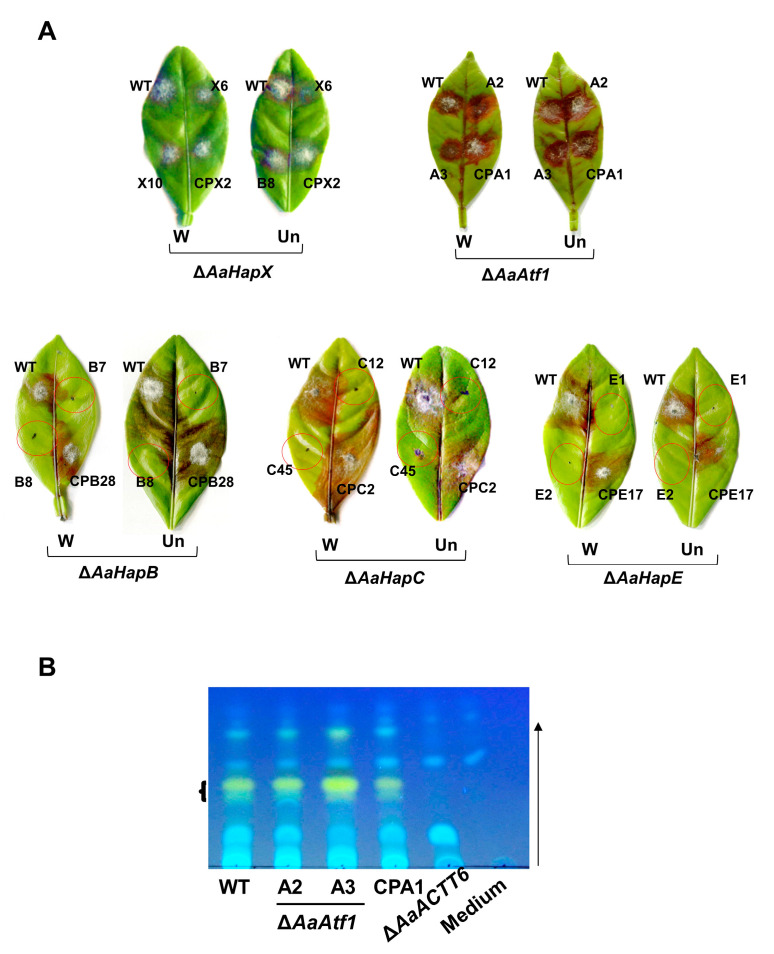
AaHapX and AaAtf1 play no role in *A. alternata* pathogenesis. (**A**) Pathogenicity tests assayed on detached calamondin leaves revealed that ∆*AaHapX* (X6 and X10) and ∆*AaAtf1* (A2 and A3) induced necrotic lesions at rates and magnitudes similar to the wild type and their respective complementation (CP) strains in pre-wounded (W) or un-wounded (Un) leaves. In contrast, ∆*AaHapB* (B7 and B8), ∆*AaHapC* (C12 and C45), and ∆*AaHapE* (E1 and E2) barely induced necrotic lesions. (**B**) TLC analysis of toxin purified from wild type, ∆*AaAtf1* (A2 and A3), and the complementation strain (CPA1) revealed that deleting *AaAtf1* had no effects on ACT production. No ACT was detected from medium only or culture filtrates of ∆*AaACTT6* (an ACT deficient strain). The arrow indicates the direction of the mobile phase (solvent).

**Figure 8 jof-09-00427-f008:**
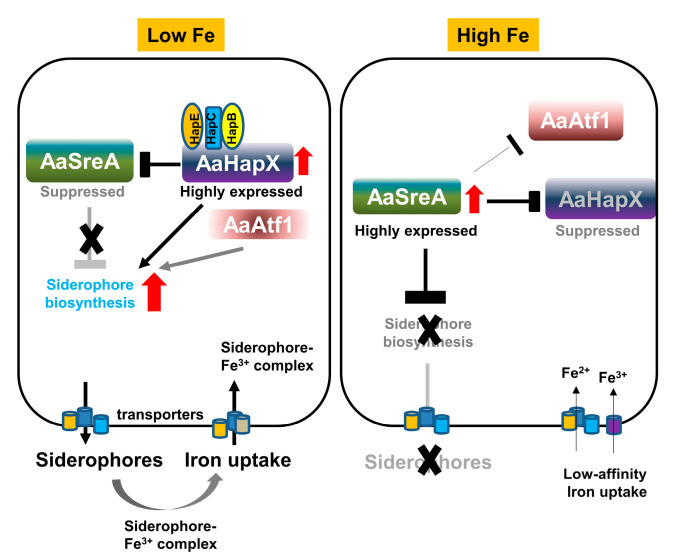
Schematic diagram illustrating the interaction of AaSreA, AaHapX, three CCAAT-binding proteins (AaHapB, AaHapC, and AaHapE), and AaAtf1 in relation to siderophore biosynthesis and iron uptake in *A. alternata*. Under low-iron conditions, the expression of *AaHapX* was upregulated. AaHapX was recruited to bind to the AaHapB/AaHapC/AaHapE complex, which in turn suppressed the expression of *AaSreA* encoding a repressor against siderophore biosynthesis. Without AaSreA, the genes involved in siderophore biosynthesis, transport, and iron uptake were de-repressed. As such, siderophores were produced to chelate iron from the environment. AaAtf1 enhanced the expression of *AaNps6*, thus playing a positive role in siderophore production. Under high-iron conditions, the expression of *AaSreA* was upregulated, which in turn suppressed the expression of *AaHapX*, *AaAtf1*, and genes involved in siderophore biosynthesis, thus preventing siderophore production.

## Data Availability

The data presented in this study have been deposited at NCBI under BioProject accession PRJNA666299.

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
