# Peer review of "The Regulatory Hub of Siderophore Biosynthesis in the Phytopathogenic Fungus Alternaria alternata"

_jof, 2023, doi:10.3390/jof9040427_

Round 1

Reviewer 1 Report

Wu et al. sought to decipher the transcriptional loop involved in iron acquisition in Alternaria alternate. Authors performed a series of functional analyses of transcription factors. The results indicated that all transcription factors do not only play significant roles in siderophore biosynthesis and iron homeostasis, but also are involved in fungal resistance to osmotic stress. This work is very interesting and deserves further investigation, which will facilitates understanding of iron acquisition in the pathogenic fungi. The manuscript was written well and strongly recommended to the journal JoF.

Major suggestions:

1. In section 3.6. It is better to provide the data not shown as the supplementary information.

Minor suggestions:

NO.

Reviewer 2 Report

The authors investigated the complex regulatory network of iron homeostasis in A. alternata. Similar biological roles of HapX and SreA reported in other fungi have been validated in A. alternata, and new regulator, Atf1, was found to be involved in siderophore production. 

I only have few comments:

1) transcriptome or proteome analysis could be performed to further support the relationship between the key genes, and explore new target genes of regulators.

2) Can the author explain the foundation/reason of selecting Atf1 as a candidate gene for regulating siderophore production?

3) The exact interaction between different regulators and their targets need to be further verified by more direct evidences, such as Chip-Seq or protein-interaction analysis. The authors have to avoid oversimplify their results/model, and need to discuss other possible genes/interaction that could involve in the regulatory network. 

Round 2

Reviewer 2 Report

I have no further comments